# Affinity-Point Graph Convolutional Network for 3D Point Cloud Analysis

**Yang Wang * and Shunping Xiao**

College of Electronic Science, National University of Defense Technology, Changsha 410073, China; spxiao@nudt.edu.cn
* Correspondence: wangyangs4@nudt.edu.cn

**Abstract:** Efficient learning of 3D shape representation from point cloud is one of the biggest requirements in 3D computer vision. In recent years, convolutional neural networks have achieved great success in 2D image representation learning. However, unlike images that have a Euclidean structure, 3D point clouds are irregular since the neighbors of each node are inconsistent. Many studies have tried to develop various convolutional graph neural networks to overcome this problem and to achieve great results. Nevertheless, these studies simply took the centroid point and its corresponding neighbors as the graph structure, thus ignoring the structural information. In this paper, an Affinity-Point Graph Convolutional Network (AP-GCN) is proposed to learn the graph structure for each reference point. In this method, the affinity between points is first defined using the feature of each point feature. Then, a graph with affinity information is built. After that, the edge-conditioned convolution is performed between the graph vertices and edges to obtain stronger neighborhood information. Finally, the learned information is used for recognition and segmentation tasks. Comprehensive experiments demonstrate that AP-GCN learned much more reasonable features and achieved significant improvements in 3D computer vision tasks such as object classification and segmentation.

**Keywords:** 3D point cloud analysis; deep learning; graph convolution network; 3D classification; semantic segmentation





## 1. Introduction

With the developments of 3D acquisition equipment, 3D data have played an important role in practical applications. As the foundation of current point cloud analysis algorithms, 3D shape representation learning is important for tackling 3D computer vision tasks, including 3D reconstruction [1,2], shape synthesis and modeling [3,4], 3D object classification and segmentation [5–7], as well as graphics applications such as virtual avatars [8,9].

Since Convolutional Neural Networks (CNNs) have achieved great success in numerous areas, recent research has attempted to extend it from the regular date domain (such as image, voice signal, and video data) to unorganized 3D point clouds [6–13]. However, it is challenging to apply CNN directly to 3D geometric data since they do not possess a regular Euclidean structure. A regular receptive field is required for the standard convolution kernels to extract features, since the convolution weights are fixed at a specific position of the convolution area. To solve this problem, regularization methods rasterize the irregular point clouds into probabilistic occupancy in 3D space [14–16] or project it into several 2D images [12,13]. However, this kind of transformation always introduces loss of information during the quantization procedure. Meanwhile, this kind of position-determined weight results in isotropy of the convolution kernel to the features of neighboring points.

Inspired by the great success of graph neural networks (GNNs) in other fields, some researchers tried to use it to handle 3D point cloud tasks [17–21]. They represented 3D

point clouds as graphs, in which the number and orientation of each node's neighbors varied from one to another (node inconsistency). Then, an effective convolution operation similar to the traditional operation was defined, which was defined on Euclidean structured data for 3D shape representation learning. Many graph convolutional neural networks have been developed to deal with irregular point cloud data and have achieved promising results. However, they still obtain isotropic features, as they simply and roughly connect the neighbor to form the graph structure.

To solve this problem, we designed a method that was motivated from the idea that the feature can be used to shield partial convolution weights according to the characteristics of the neighbors. Therefore, the actual receptive field of a point cloud convolution kernel is no longer a regular three-dimensional box but a shape that dynamically adapts to the structure of the object. In addition, as an important part of the graph, edges are used to enrich the features obtained.

In summary, the contributions of this paper are as follows:

(1) Affinity-Point Graph Convolutional Network (AP-GCN) is proposed for 3D point cloud analysis, which contains a graph neural network with a trainable neighbor structure. The neighbor selection method is used to replace the common spherical neighborhood and the k-NN (K-Nearest Neighbor) method.

(2) Edge-conditioned convolution is designed according to the characteristics of message transmission on graph network.

(3) A comparative 3D object classification accuracy is achieved based on the Model-Net40 classification benchmark [14] and the ShapeNet part benchmark [22].

## 2. Related Works

In this section, related studies are discussed within two main aspects: deep learning on point clouds and convolution on graphs.

### 2.1. Deep Learning on Point Clouds

With the rapid development of deep learning, CNNs have been successfully used in 2D images and researchers have tried to explore its feature learning power for 3D point clouds. However, the implementation of typical convolutions relies on a regular data format that is not satisfied in point clouds. To solve this problem, various methods have been proposed.

A straightforward extension is a voxel-based method that transforms the irregular point clouds into regular volumetric distribution [14–16]. Voxels are similar to the pixels in 3D space instead of 2D images. Then, the typical 3D convolution operation can be applied to extract its features. Wu et al. [14] transformed the point clouds into binary occupancy of voxels, where 1 stands for the occupancy and 0 stands for empty. Maturana and Scherer [15] proposed three different representations of voxels and introduced a network named VoxNet for classification tasks. VoxNet can be trained by typical SGD algorithms. Though simple, this kind of method induces a loss of information, excessive consumption of memory, and a high computation cost.

Another solution is multi-view-based methods [12,13]. They used multiple 2D images to represent 3D objects so that it can benefit from these well-developed recognition methods in 2D images [23]. Su et al. [12] generated many projection images for 3D objects by placing several virtual cameras arounds it. Traditional networks on images such as VGGNet were utilized to extract features from each individual image. The extracted features were fused subsequently to obtain the overall representation of the object. Yang et al. [13] introduced a relation network to learn the relation among different views; then, the features from all viewpoints were aggregated to produce a global feature. However, the determination of the number of 2D images is still a problem, and these methods would lead to information loss during the projection process.

Pioneered by PointNet [6], point-based methods directly consume point clouds rather than transform them into an intermediate representation. An effective and simple archi-

tecture was constructed in PointNet to directly learn from point sets. First, point features wer computed individually from shared multi-layer perceptrons (MLPs). Then, all of the features were aggregated as a global presentation of a point cloud. Furthermore, Point-Net++ [7] was proposed to incorporate the local information since PointNet processes all points at one time. Although these methods significantly improve the speed of the 3D shape analysis, they still ignore structural information.

### 2.2. Convolution on Graphs

The graph-based methods [17–21] generalize the standard CNNs to adapt to the graph-structural data by representing the point cloud as a graph according to their spatial neighbors.

Convolution on graphs includes spectral approaches and non-spectral approaches. The spectral approaches define convolutions as spectral filtering, which is implemented as the multiplication of signals on a graph with eigenvectors of the graph Laplacian matrix [24,25]. Defferrard et al. [24] proposed truncated Chebyshev polynomials to approximate the spectral filtering. The learned feature maps are located within the K-hops neighbors of each point. RGCNN [26] constructs a graph by connecting each point with all other points in the point cloud, updating the graph Laplacian matrix in each layer. To make features of adjacent vertices more similar, a graph-signal smoothness is added a priori into the loss function. Wang et al. [27] proposed an end-to-end spectral convolution network LocalSpecGCN to work on a local graph (which is constructed from the k nearest neighbors). This method does not require any offline computation of the graph Laplacian matrix and graph coarsening hierarchy. However, the spectral convolution still suffers from a high computation cost, and it cannot be transferred to another graph since they have different Laplacian matrices.

The key to the non-spectral methods is that they directly define the convolution on graphs with local neighbors in spatial or manifold domains. Specifically, convolution is usually implemented using MLP over spatial neighbors. Pooling produces a new coarsened graph by aggregating information from each point's neighbors. Simonovsky et al. [28] considered each point as a vertex of the graph and connected each vertex to all its neighbors using a directed edge. Then, Edge-Conditioned Convolution (ECC) was proposed using a filter-generating network (e.g., MLP). Max pooling was adopted to aggregate neighborhood information, and graph coarsening was implemented based on the VoxelGrid [29] algorithm. Liu et al. [30] proposed a Dynamic Points Agglomeration Module (DPAM) based on graph convolution to simplify the process of point agglomeration (sampling, grouping, and pooling) into a simple step, which was implemented through multiplication of the agglomeration matrix and points feature matrix. Duvenaud et al. designed a weight matrix for each vertex and multiplied it to its neighbors, following a sum operation [31]. Niepert et al. [32] selected and sorted the neighbors of each vertex, so that 1D CNNs can be used.

At present, most methods performed neighborhood feature extraction with a fixed neighborhood structure and size (such as spherical shapes, squares, etc.). However, the neighborhood structure of a target always suffers from multiple scales and has no fixed shape. As a result, the traditional methods cannot effectively obtain complete information from a fixed neighborhood. In view of this, a graph-based method is proposed to obtain a much more reasonable neighborhood and its feature. Furthermore, a neighborhood construction method is designed to aggregate neighbor information from both vertices and edges in the built graph. This construction is useful for capturing neighbors with stronger affinity to the neighborhood center. Additionally, based on the built graph and neighborhood structure, a convolution operation is proposed to learn features for the recognition task.

## 3. Method

First, a novel neighbor selection method was proposed for structured feature learning of 3D point cloud. After that, an end-to-end 3D point cloud classification and segmentation framework was constructed.

### 3.1. Neighbor Selection Method

The most popular neighbor selection method is k-NN. The core idea of the k-NN method is to traverse all points and then to select the neighbor according to the distance. A spherical neighbor is divided by the azimuth angle, the pitch angle, and the radius. The essence of these methods is to select a neighbor based on the Euclidean distance.

Considering a graph constructed from a given point cloud $P = \{p_1, p_2, \ldots, p_N\} \in R^3$, which is described as a mesh $G = (V, E)$, where $V = \{v_1, v_2, \ldots, v_N\}$ is a set of vertices and $E \subseteq |V| \times |V|$ is a set of edges. $N$ is the number of vertices (points). The neighbor set of vertex $p_i$ is denoted as $N(i) = \left\{ p_i, p_{i,1}, \ldots, p_{i,|N(i)|} \right\}$, where $|N(i)|$ is the number of the neighbors of $p_i$. It is worth noticing that this set contains itself. Let $h = \{h_1, h_2, \ldots, h_N\}$ be the set of input vertex features and $h_i \in R^F$ be the feature of $p_i$, where $F$ is the feature dimension.

The proposed method is designed to learn a function $\phi : R^3 \rightarrow R^1$, which is used to measure the affinity between neighbors and the center point. First, the distance between neighbors is obtained as $p_{ij}^* = p_i^* - p_j$, where $p_i^* \in P$ is the center point and $p_j \in P$ represents its neighbor. Then, the specific values are used to evaluate the affinity between the neighbor and its center, as follows:

$$\alpha(i, j) = \phi\left( p_{ij}^*, w_i \right) \tag{1}$$

where $w_i \in R^3$ is a trainable weight vector of $p_i^*$. $\alpha(i, j)$ representing the affinity of $p_j$ to $p_i^*$.

In this paper, the operation $\phi : R^3 \rightarrow R^1$ is defined by multiplication, so that Equation (1) can be written as follows:

$$\alpha(i, j) = p_{ij}^* \bullet w_i^T \tag{2}$$

where $\alpha(i, j)$ is a constant.

In addition, in order to deal with a neighborhood with different scales, the affinity value is normalized across all of the neighbors of vertex $p_i^*$, as follows:

$$\alpha'(i, j) = \frac{\exp(\alpha(i, j))}{\sum_{l \in N(i)} \exp(\alpha(i, l))} \tag{3}$$

Thus, an affinity ranking for vertex $p_i^*$ can be obtained and the top $k$ points is manually selected. The chosen $k$ neighbors is represented as $N'(i) = \left\{ p_i^*, p_{i,1}^*, \ldots, p_{i,k}^* \right\}$. Additionally, the edge direction information is represented using $\alpha'(i, j)$ and $\alpha'(j, i)$. If $\alpha'(i, j) > 0$ and $\alpha'(j, i) > 0$, the points are doubly linked. In the other situation, the points are one-way linked. Directed edges between a vertex and its neighbors are shown in Figure 1.

As shown in Figure 1, $p_i^*$ and $p_4$ are doubly linked, which means that they have a strong affinity for each other and information transmission between them goes two ways. $p_i^*$ and $p_1$ are one-way linked, which means that $p_i^*$ has a strong affinity to $p_1$ and information transmission between them is goes only one way. Obviously, the key to the use of this graph is to learn information about both the vertices and the edges.

### 3.2. Convolution with Vertices and Edges

As defined above, both the edges and the vertices are important parts of a graph. Moreover, edges contain rich structural information. Therefore, a convolution operation is needed to capture local information from the vertices and structure information from the edges. A general convolutional operation is formulated as follows:

$$h' = \sigma\left( A\left( \Gamma\left( \alpha'(i, j), \alpha'(j, i), h_j \right) \right) \right) \tag{4}$$

where $h'$ is the final output feature. $\Gamma\left(\alpha'(i,j), \alpha'(j,i), h_j\right)$ is used to obtain local information with vertex feature $h_j$ and structural information with affinity information $\alpha'(i,j)$ and $\alpha'(j,i)$. $\alpha'(i,j)$ represents the information between vertices $p_i^*$ and $p_j$, while $\alpha'(j,i)$ represents the information between vertices $p_j^*$ and $p_i$. In detail, $\Gamma(\bullet)$ is implemented with a shared multi-layer perceptron (MLP). After the MLP layers, a feature fusion method $A$ and a nonlinear activator $\sigma$ follow. To maintain permutation invariance of the point set, $A$ is set as a symmetric function (such as max pooling) and $\Gamma(\bullet)$ is shared over each neighbor.

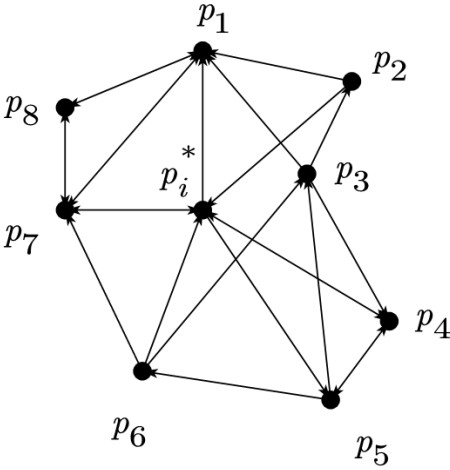

**Figure 1.** An example of a learned graph structure.

### 3.3. Graph Pyramid Construction on a Point Cloud

In this section, we explain how to construct a graph pyramid on point clouds according to their neighbors. Many studies directly search for the spatial neighbors of all points and link them as a graph. After that, a graph pyramid with different spatial scales is constructed by alternately applying graph construction and coarsening techniques.

**Graph construction on a point cloud.** Given a point cloud $P$, which is a set of the spatial coordinates of the points, a directed graph $G = (V, E)$ is constructed. Each vertex $v_i$ is associated with a center point $p_i^*$, and the edges are added between the point and its $K_l$ neighbors $N'(i) = \left\{ p_i^*, p_{i,1}^*, \ldots, p_{i,K_l}^* \right\}$. The direction from neighbor points to the vertex represents the direction of information transmission. In our experiments, the $K_l$ neighbors are learned from its features, which performs better than random sampling.

**Graph coarsening.** To reduce the computation cost and memory consumption, the input point cloud $P$ is subsampled with a set of ratios using the furthest point sampling algorithm. Then, a corresponding graph $G$ can be constructed as described above.

**Graph pooling.** To aggregate the information from neighboring vertex to the centroid vertex, graph pooling is a necessary procedure. For each local graph, the operation of graph pooling is used to aggregate the vertex features. In this paper, the max pooling adopts a pooling function. The specific definitions are as follows:

$$h_i^{l+1} = pooling\left\{ h_j^l : j \in N_l(i) \right\} \tag{5}$$

where $N_l(i)$ is the neighbor set of the $l$-level vertex $v_i$ and $h_j^l$ is the corresponding feature of the neighbors. $h_i^{l+1}$ is the output feature of the pooling operations.

**The network architecture.** The architecture of AP-GCN is shown in Figure 2. The subsampled point cloud is directly used as the input, and the directed graph is constructed using the proposed method. Then, the convolution operation is performed on the vertices and edges. After each coarsening, the number of neighbors decreases. For the segmentation task, the main structure is the same as the classification network. Different from the classification network, feature interpolation is necessary to obtain the feature mapping

to each point. As a result, the learned features are restored from the coarsest scale to the original scale. At the same time, different-level features are fused with a skip connection.

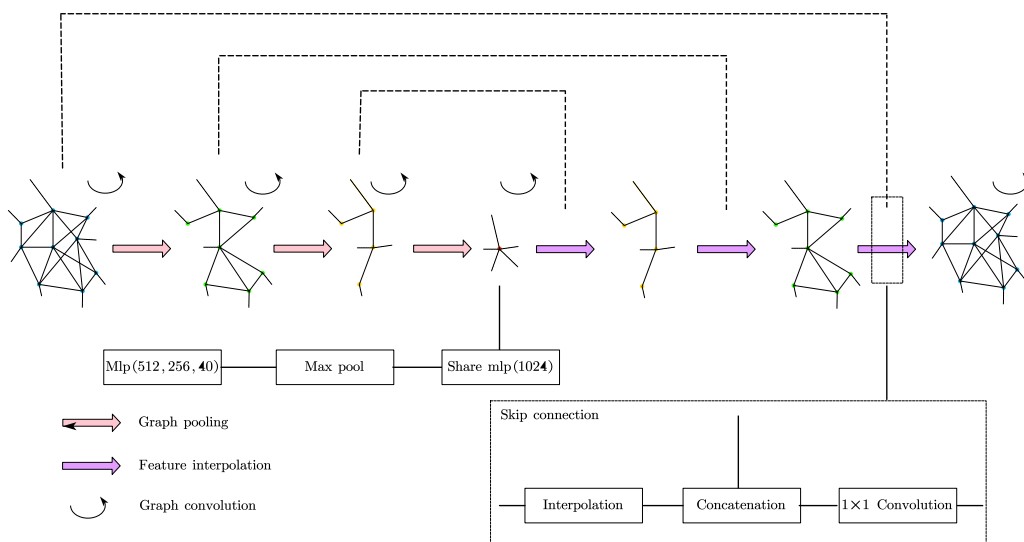

**Figure 2.** The architecture of AP-GCN.

The proposed network is generally consistent with the common recognition and segmentation architecture. The main difference lies in the construction of the graph structure and convolution operation. Our graphs are constructed by calculating the affinity between points, and the convolution operation captures both vertex and edge features.

## 4. Experiments

In this section, the proposed AP-GCN is evaluated based on the ModelNet40 classification benchmark and the ShapeNet part benchmark [14] to verify its effectiveness in classification and segmentation tasks, respectively. All of the experiments are run on a GTX 1070 GPU with CUDA 11.3 and CuDNN 7.6.5.

### 4.1. Classification on the ModelNet40 Dataset

The ModelNet40 classification benchmark consists of 9843 train shapes and 2468 test shapes in 40 categories. Here, the point cloud data provided by [6] are used for classification, 1024 points are uniformly sampled from each shape, and all sampled points are normalized to a unit sphere. Similar to [33], during training, a random anisotropic scaling with a range of $[-0.66, 1.5]$ and a translation with a range of $[-0.2, 0.2]$ were used for data augmentation.

The Adam optimizer with a momentum value of 0.9 was used for training. A dropout rate of 50% was set at each fully connected layer. Furthermore, the initial learning rate was set to 0.001, and the decay rate was set to 0.7 in all experiments During training, an early stop strategy was introduced based on the validation accuracy. The update was stopped when the accuracy did not improve after 10 epochs or the number of training epochs was more than 100. In addition, overall accuracy (OA) was used to evaluate the methods.

The quantitative comparisons with the state-of-the-art methods are summarized in Table 1. It can be seen from Table 1 that the proposed method performs better than most other methods. Even using only xyz information as the input, the proposed method achieved a comparative result (92.1%) to most of the existing models. In the same case, AP-GCN achieved results with only 0.001% lower recognition than DGCNN [19], which achieved the best recognition result. Additionally, AP-GCN achieved 1.5% higher than the classical PointNet++ [7] and 0.4% higher than the classical PointCNN [10].

The computational complexity of each method with the same input (1k points from ModelNet40) is shown in Table 1. Except PointNet, the time consumption of the proposed

method is minimal compared with the other methods. It can be seen that the proposed also achieved the lowest computational complexity.

**Table 1.** Shape classification results (%) based on the ModelNet40 benchmark.

| Method | Input | Points | OA (%) | Time Consumption (ms) |
|---|---|---|---|---|
| Pointwise-CNN [34] | xyz | 1k | 86.1 | 6985 |
| PointNet [6] | xyz | 1k | 89.2 | 189 |
| PointNet++ [7] | xyz | 1k | 90.7 | 8761 |
| PointCNN [10] | xyz | 1k | 91.7 | 5136 |
| Ours | xyz | 1k | 92.1 | 4218 |
| DGCNN [19] | xyz | 1k | 92.2 | 5562 |
| SO-Net [11] | xyz | 2k | 90.9 | - |
| Kd-Net (depth = 15) [33] | xyz | 32k | 91.8 | - |

*4.2. Part Segmentation on the ShapeNet Dataset*

Compared with classification, part segmentation is a much more challenging task. The proposed method was evaluated on the ShapeNet dataset [22]. ShapeNet dataset is composed of 16,881 shapes with 16 categories, which are labeled with 50 parts in total. Following the data splitting operation in [6], 2048 points were randomly picked as the input of the proposed network. Furthermore, the part segmentation performances were compared using the average category mIoU and the average instance mIoU.

For target recognition, the set of points with real true labels was assumed to be $A$, and the set of points marked by the algorithm as the $A$ type was $B$. Then, IoU can be calculated as follows:

$$IoU = \frac{[A \cap B]}{[A \cup B]} \tag{6}$$

where $[A]$ denotes the number of points in $A$. The average category mIoU describes the mean *IoU* of the overall classes, while the average instance mIoU describes the mean *IoU* of the overall instances.

The results of shape part segmentation on rotated shapes are shown in Table 2. AP-GCN achieves a class mIoU of 83.6% and an instance mIoU of 85.4%, which are both optimal results. The class mIoU increases by 1.3% compared with DGCNN, which obtained the second best result. Compared with the best additional-input method, AP-GCN is obviously better than PiontNet++ [7] and SO-Net [11], while AP-GCN only takes an xyz coordinate as input. These results showed that AP-GCN can capture more accurate structural information because of the addition of both vertices and edges. Figure 3 shows some examples of part segmentation. It can be seen that AP-GCN can segment parts correctly.

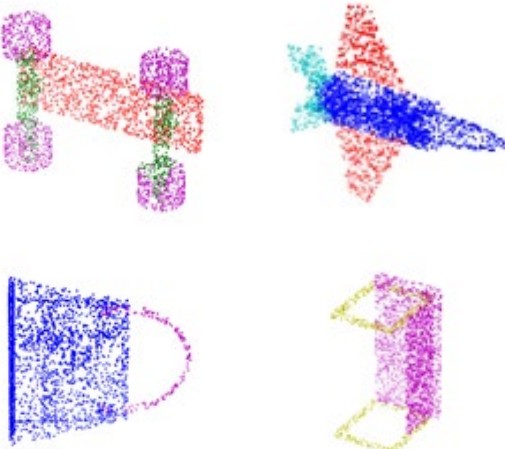

**Figure 3.** Several part segmentation results obtained by AP-GCN.

**Table 2.** Shape part segmentation IoU results (%) on the ShapeNet dataset. The best results are marked in bold.

| | Input | Class mIoU | | | | Instance mIoU | | |
|---|---|---|---|---|---|---|---|---|
| PointNet | 2k | 80.4 | | | | 83.7 | | |
| DGCNN | 2k | 82.3 | | | | 85.1 | | |
| Kd-Net | 4k | 77.4 | | | | 82.3 | | |
| Ours | 2k | **83.6** | | | | **85.4** | | |
| SO-Net | 1k + norm | 80.8 | | | | 84.6 | | |
| PointNet++ | 2k + norm | 81.9 | | | | 85.1 | | |
| | **Input** | **Aero** | **Bag** | **Cap** | **Car** | **Chair** | **Guitar** | **Knife** |
| PointNet | 2k | 83.4 | 78.7 | 82.5 | 74.9 | 89.6 | **91.5** | 85.9 |
| DGCNN | 2k | **84.2** | 83.7 | 84.4 | 77.1 | 90.9 | **91.5** | 87.3 |
| Kd-Net | 4k | 80.1 | 74.6 | 74.3 | 70.3 | 88.6 | 90.2 | 87.2 |
| Ours | 2k | 83.4 | **84.7** | **87.8** | **79.3** | **91.0** | 91.0 | **88.1** |
| SO-Net | 1k + norm | 81.9 | 83.5 | 84.8 | 78.1 | 90.8 | 90.1 | 83.6 |
| PointNet++ | 2k + norm | 83.5 | 81.0 | 87.2 | 77.5 | 90.7 | 91.1 | 87.3 |
| | **Input** | **Lamp** | **Laptop** | **Motor** | **Mug** | **Pistol** | **Rocket** | **Table** |
| PointNet | 2k | 80.8 | 95.3 | 65.2 | 93.0 | 81.2 | 57.9 | 80.6 |
| DGCNN | 2k | 82.9 | **96.0** | 67.8 | 93.3 | 82.6 | 59.7 | 82.0 |
| Kd-Net | 4k | 81.0 | 94.9 | 57.4 | 86.7 | 78.1 | 51.8 | 80.3 |
| Ours | 2k | **84.1** | 95.8 | **73.7** | 94.1 | **83.0** | **60.4** | **83.3** |
| SO-Net | 1k + norm | 82.3 | 95.2 | 69.3 | **94.2** | 80.0 | 51.6 | 82.6 |
| PointNet++ | 2k + norm | 83.3 | 95.8 | 70.2 | 93.5 | 82.7 | 59.7 | 82.8 |

Additionally, it should be noted that AP-GCN lag behind other methods in the aero, guitar, laptop, and mug classes. Compared with other classes, these four classes suffer much more small structural parts. Due to this reason, AP-GCN may miss some local features. In view of this, we need to add more local information extraction structures to AP-GCN.

### 4.3. Ablation Experiments

In this section, additional experiments are conducted to evaluate the effectiveness of the proposed neighbor selection method. Take the recognition task as an example; the experiments are carried out under a similar network with a different neighborhood selection method. To be clear, the number of neighborhoods is consistent ($K = 100$). The classification results on the ModelNet40 dataset are shown in Table 3. As we can see from Table 3, the proposed method achieved the best results. The results demonstrated that the proposed neighbor selection method can obtain a much better neighborhood for feature learning.

**Table 3.** Classification results on ModelNet40 with different neighbor selection methods.

| Neighborhood Selection Method | OA (%) |
|---|---|
| k-NN | 91.7 |
| Ball query | 92.0 |
| Ours | 92.1 |

### 4.4. Discussion of AP-GCN

Due to the two modules of learnable neighbor selection mechanisms, and vertex- and edge-based convolution operation, the proposed AP-GCN outperforms the classical approaches. Compared with classical neighbor selection methods such as farthest point sampling or random sampling according to a certain radius, the proposed neighbor selection method can learn much more information. In detail, the proposed method fully considers the relationship between each point and can construct a graph structure that can

reflect the actual relationship more accurately based on affinity measurements. On the other hand, the proposed convolution operation based on graph vertices and edges captures much more structure information. As an important part of the graph, the edge in a graph contains important structure information between the vertexes in the graph. The full use of edges and vertices can better contribute to multi-scale structural information extraction.

## 5. Conclusions

In this paper, the Affinity-Point Graph Convolutional Network (AP-GCN), which can be used to learn graph structures on point cloud and to capture the edge structural features, was proposed for 3D point cloud analysis. The core components of AP-GCN are the affinity measurement method and the convolution operation, which can combine edge and vertex features. From the affinity information, a graph can be adaptively extracted from a point cloud. Additionally, an efficient feature learning is performed using the graph and the proposed convolution operation. The experimental results show the effectiveness of AP-GCN, especially its best results achieved in the part segmentation task. Although the AP-GCN achieves success, the proposed affinity measurement is just a simple method with a Euclidean distance, and a much more efficient method is needed in future research.

**Author Contributions:** Conceptualization, Y.W. and S.X.; methodology, Y.W. and S.X.; software, Y.W.; validation, Y.W. and S.X.; formal analysis, Y.W. and S.X.; investigation, Y.W.; resources, Y.W.; data curation, Y.W.; writing—original draft preparation, Y.W.; writing—review and editing, Y.W. and S.X.; visualization, Y.W. and S.X.; supervision, S.X.; project administration, Y.W. and S.X. All authors have read and agreed to the published version of the manuscript.

**Funding:** This research received no external funding.

**Institutional Review Board Statement:** Not applicable.

**Informed Consent Statement:** Not applicable.

**Data Availability Statement:** Not applicable.

**Conflicts of Interest:** The authors declare no conflict of interest.

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
