# Peer review of "Affinity-Point Graph Convolutional Network for 3D Point Cloud Analysis"

_applsci, doi:10.3390/app12115328_

Round 1

Reviewer 1 Report

The paper propose a point clouds classification and segmentation approach using graph neural network is proposed. the cloud analysis, which contains a graph neural network with trainable neighbor structure is presented.  A comparative 3D object classification accuracy is achieved on the ModelNet40. The study has shown significant improvement over the classical approaches. however, the following minor issues should be addressed

  1.  Give possible reason why the propose approach outperform the classical approaches.
  2.  Ensure all the variables in equation 4 are defined
  3. Improve the visual quality of figure 4 
  4. show percentage improvement of your approach over the classical methods
  5. Removed all published not peer review references in the paper, eg ref 40
  6. Add 2022 papers for update 

Reviewer 2 Report

The paper presents a novel Graph CNN for 3D point cloud analysis by learning affinity between 3D points. The idea presented in novel and intuitive, however the paper needs improvement for it to be published in the journal. Some of my observations and comments which can be applied in the paper are as follows:

  • English and grammar correction - Several places, the use of language would not help readers. For example, ". In this method, the neighborhood 
    17 is learned according to a certain rule---the affinity between points, to get stronger neighborhood 
    18 points and than the edge-conditioned convolution(ECC) is implemented".
  • The Section 2.2 does not talk about the the reason why new convolution operation is designed for the given task. The discussion should highlight the novelty in APGCN vis-a-vis other methods, in particular with PointNet.
  • The Section 3.1 should specify how the graph stores the affinity information between different points. Further, the authors should also discuss how the edge direction information is stored in the graph structure.
  • The discussion on the learning objective and training parameter selection is grossly missing which is required for complete understanding of the proposed method.
  • The conclusion section does not discuss about the possibilities of extension in this work. The authors should also include discussion on computational complexity of the proposed method for completeness sake.

Incorporation of above listed points would certainly improve the overall quality of paper.

Reviewer 3 Report

  1. Paper description

This paper presents a Graph neural network-based 3D point cloud processing method for classification and part segmentation, which uses the affinity information between a point and its neighbour for neighbour selection. This work proposes a novel neighbour selection method to replace the common spherical neighbourhood and K- Nearest Neighbour method. Also, proposed method uses both vertex and edge information for feature extraction.

  1. Strengths

Reviewer feels that the paper provides architectural and experimental details. The novelty of this work lies in the affinity measurement method for neighbour selection and utilization of edge information along with vertex information in convolution operation. Experimental results of part segmentation on Shapenet dataset shows improvement over state-of-the-art.

  1. Weakness
  • Contribution 1 (line 58-59) seems redundant and is already covered in other mentioned contributions.
  • Line 87-96 is irrelevant in the context of Deep learning methods on point clouds. Related works should include more relevant works.
  • In line 157, it is not clearly described that why only |N(i)|-1 neighbours are considered in neighbour set of vertex i, not |N(i)|. Similarly in line 177, why kth neighbour is not considered.
  • In line 172-173, sentence “….in order to deal with the neighbour with size change on different vertices and spatial scales,….” is not clear. Elaborate more about neighbour with size change.
  • Describe more about the contribution of directed edges. Also, on what basis these directions are assigned to the edges between vertices.
  • In Fig. 2, more description of the architecture in required.
  • Performance comparison of proposed method against state-of-the-art for classification does not provide proper evidence to show superiority of proposed method over them. As mentioned in line 248, proposed method obtains lesser overall accuracy against DGCNN. Therefore, a proper description or more ablation studies are required to show that how AP-GCN is better than SOTA.
  • Provide explanation to support why AP-GCN lacks behind other works in 4 categories in Table-2.
  • In ablation experiments, what is the number of neighbors K taken for experiments?
  • In discussion, “compared with taking neighbours….. “ is not clear (refer line 290-292).
  • There are several grammatical and sentence formation errors and misspellings throughout the whole paper which sometimes create ambiguity in describing the content/theory (Refer lines 18, 26, 32 44, 47-48, 52, 55-56, 58, 76, 80, 87,88, 90, 99, 131, 176, 187-188, 214-215, 217, 219, 225, 254, 255, 264, 299-300, 306-307, 419-420 and word “PointNet” and “pointwise” in Table 1).
  • Proper citation is missing (refer line 97, 137, 139).
  • Table 2 is not aligned properly.

  1. Recommendation

This work seems to have its merits. However, reviewer feels that better exposition, formatting and elaboration is required. Authors should resolve all the grammatical and sentence formation mistakes in the paper to avoid the obscurity for readers. Also, authors should address all the other queries and provide proper explanation for each of them.
